# In-situ liquid cell transmission electron microscopy investigation on oriented attachment of gold nanoparticles

Chao Zhu[1], Suxia Liang[2], Erhong Song[3], Yuanjun Zhou[4], Wen Wang[1], Feng Shan[5], Yantao Shi[2], Ce Hao[2], Kuibo Yin[1], Tong Zhang[5], Jianjun Liu[3], Haimei Zheng[6] & Litao Sun [1,7,8]

Inside a liquid solution, oriented attachment (OA) is now recognized to be as important a pathway to crystal growth as other, more conventional growth mechanisms. However, the driving force that controls the occurrence of OA is still poorly understood. Here, using in-situ liquid cell transmission electron microscopy, we demonstrate the ligand-controlled OA of citrate-stabilized gold nanoparticles at atomic resolution. Our data reveal that particle pairs rotate randomly at a separation distance greater than twice the layer thickness of adsorbed ligands. In contrast, when the particles get closer, their ligands overlap and guide the rotation into a directional mode until they share a common {111} orientation, when a sudden contact occurs accompanied by the simultaneous expulsion of the ligands on this surface. First-principle calculations confirm that the lower ligand binding energy on {111} surfaces is the intrinsic reason for the preferential attachment at this facet, rather than on other low-index facets.

[1] SEU-FEI Nano-Pico Center, Key Laboratory of MEMS of Ministry of Education, Collaborative Innovation Center for Micro/Nano Fabrication, Device and System, Southeast University, Nanjing, 210096, China. [2] State Key Laboratory of Fine Chemicals, School of Chemistry, Dalian University of Technology, Dalian, 116024, China. [3] The State Key Laboratory of High Performance Ceramics and Superfine Microstructure, Shanghai Institute of Ceramics, Chinese Academy of Sciences, Shanghai, 200050, China. [4] Department of Physics and Astronomy, Rutgers, The State University of New Jersey, Piscataway, NJ 08854, USA. [5] Joint International Research Laboratory of Information Display and Visualization, School of Electronic Science and Engineering, Southeast University, Nanjing, 210096, China. [6] Materials Sciences Division, Lawrence Berkeley National Laboratory, Berkeley, CA 94720, USA. [7] Center for Advanced Carbon Materials, Southeast University and Jiangnan Graphene Research Institute, Changzhou, 213100, China. [8] Center for Advanced Materials and Manufacture, Joint Research Institute of Southeast University and Monash University, Suzhou, 215123, China. Correspondence and requests for materials should be addressed to H.Z. (email: hmzheng@lbl.gov) or to L.S. (email: slt@seu.edu.cn)

O riented attachment (OA), proposed by Peen and Banfield about two decades ago[1], is now widely accepted as a viable route for the synthetic production of various nanomaterials ranging from quantum dots to one dimensional (1D) nanowires, 2D nanosheets and complex hierarchical 3D nanostructures[2–6]. Different from the canonical Ostwald ripening (OR)[7] where nanoparticles grow larger at the expense of small ones, OA refers to another growth mode in which, nearby nanoparticles with pre-aligned crystallographic orientations coalesce or aggregate into a single particle. In comparison with OR, OA mechanism has been successful in explaining the growth of irregular or anisotropic single crystalline nanostructures[8,9], as well as the formation of planar defects (stacking faults, twins, etc.)[10].

Despite its advantages, there are still many debates about the driving force that kinetically controls the occurrence of OA. Given the complexity of the environment in a solution, a number of factors need to be taken into account to elucidate their individual contributions during successive OA events[11–13]. In the view of themodynamics, reduction in surface free energy in way of the elimination of high energy surfaces is the primary driving force for spontaneous OA[14–16]. For nanocrytals possessing electric dipoles which is due to the non-centrosymmetric atomic lattice or magnetic dipoles which originates from the constituent magnetic atoms, their dipole–dipole interaction is described as the driving force to steer OA along the dipole orientation[8,17,18]. The asymmetrical distribution of electric charges on different surfaces are also assumed to induce anisotropic charge–charge driving force and thus OA at specific surfaces[19,20]. In some other systems where Coulomb interaction is screened, van der Waals force is considered as the physical driving force since it can act over long distance[21,22]. In addition to these, increasing evidence indicate that capping ligands should play a decisive role in OA related growth[23,24]. For instance, the occurrence of OA can be manipulated on different surfaces of PbSe by just changing the organic ligands in synthetic process[25]; and amine capped Au nanoparticles undergo continuous OA to form single crystalline nanowires after adding ascorbic acid into the colloidal solution[4]. However, nearly no experiment or simulation results have clarified the detailed mechanism about how ligands control OA process because the previous ex-situ transmission electron microscope (TEM) method is insufficient to obtain direct evidence.

The development of a liquid cell TEM[26,27] enables the direct observation of nanoscale chemical reactions and the physical behavior in a solution, where nucleation[28], growth[29], self-assembly[30], and dissolution[31], that have not been visualized previously, can be studied now. Recently by using this technology, it has been demonstrated that dipole–dipole interaction leads to imperfect OA of Pt₃Fe[32], and other group suggests the Coulomb interaction seems to be responsible for OA of Fe₂O₃ although van der Waals force cannot be ruled out[33].

Here, we used an ordinary carbon film based liquid cell (Supplementary Fig. 1 and 2) for in-situ high resolution TEM investigation of this mechanism in gold nanoparticles. The motion and orientation of particle pairs are tracked during their OA process. Systematic statistical analysis allows us to uncover the ligands related dynamics. Theoretical calculations confirm the different binding energies between ligands and crystal facets play the key role in preferential attachment at {111} facets.

## Results

**Observation of OA trajectories**. A stable aqueous solution containing well suspended gold nanoparticles with diameters in the range of 10−20 nm was prepared by mixing 34 mM sodium citrate aqueous solution with 0.24 mM HAuCl₄ aqueous solution. About 2 μL of the resulting solution was dropped onto a piece of formvar stabilized carbon support film and covered with another similar TEM grid. The sample was left in ambient atmosphere overnight while the two TEM grids naturally bonded due to van der Waals adhesion, ensuring the presence of water islands encapsulated by the amorphous carbon film. Atomic level resolution was achieved (Supplementary Fig. 3 and 4) during real-time observations using FEI Tian 80–300 TEM with Cs-corrector. To obtain free-moving small particles, we irradiated the sample using a steady electron beam ($\sim 4 \times 10^5$ e nm$^{-2}$ s$^{-1}$) to induce a dissolution-precipitation reaction, whereupon large gold particles were continuously consumed while smaller ones with sizes of ~2 nm were generated in solution (Supplementary Movie 1 and 2, Supplementary Note 1).

Trajectories of OA were examined by observing the coalescence events of small particles. For most of the time, particles keep moving randomly in the solution. But sometimes they are found along the viewing zone axis and despite occasional off-axis fluctuations, it is possible to identify their orientations by lattice fringes. Additionally, the projection appearance of these truncated octahedron particles is also helpful to track their trajectories (Supplementary Fig. 5). TEM image sequences extracted from Supplementary Movie 3 track the formation process of a twinned structure by OA at the {111} surfaces, as displayed in Fig. 1. It can be seen that, initially the two particles have different orientations for both their respective {111} and {100} facets. When they get closer, to a distance of about 1.3 nm, the individual particles in the pair starts to rotate in opposite directions, clockwise for the left one and anti-clockwise for the right one, until their {111} facets are perfectly aligned. Here, the left particle experiences a larger rotation angle than the right one, which may be attributed to its smaller moment of inertia owing to its slightly smaller size. Once this crystal facet matching is established, OA is accomplished via a transient jump to contact behavior at the separation of 0.7 nm. The contact at their {111} surfaces leaves this crystal facet as a twin interface since the angle between their {100} facets is about 70°. After that, the connective neck between the particles vanishes through a rapid diffusion of surface atoms, leading to the formation of a larger individual particle. A monocrystalline structure can be formed in the same way if the particles contact at aligned {111} surfaces but their {100} facets are parallel with each other (Supplementary Fig. 6, Supplementary Movie 4). In our observation, it is found that coalescence will not occur if the facets are misaligned even if the particles are close enough (Supplementary Fig. 7).

**Ligands related dynamics of OA**. Quantitative analysis of OA trajectories reveals a dynamic two-stage process for the approaching pairwise particles. Figure 2 describes in detail the changes in the separation distance $D$ between a particle pair, and in the relative angle $\theta$ between their {111} facets for the entire OA process. Before jump to contact (601.7 s), the particles, subjected to the effect of some long-range force, slowly approach each other although there are fluctuations due to Brownian motion. The separation distance changes from about 2 nm to 0.7 nm, showing a roughly linear decrease with a velocity of 0.04 nm s$^{-1}$ (Fig. 2a). At this point, the separation distance can be further divided into two stages if the changes in their relative angle are considered (Fig. 2b). At stage I (2.0–1.3 nm), $\theta$ changes randomly within the range 10–45°, indicating that there is no connection between the particle pair and that they rotate freely in the solution. However, subsequently, the rotation of the particle pair appears to change to a different mode in stage II (1.3–0.7 nm), where $\theta$ gradually decreases to near 0°. Moreover, according to more experimental results, this critical distance at which directional rotation begins is always around 1.3 nm and independent of the particle sizes (Supplementary Fig. 8 and 9). This implies that when $D < 1.3$ nm,

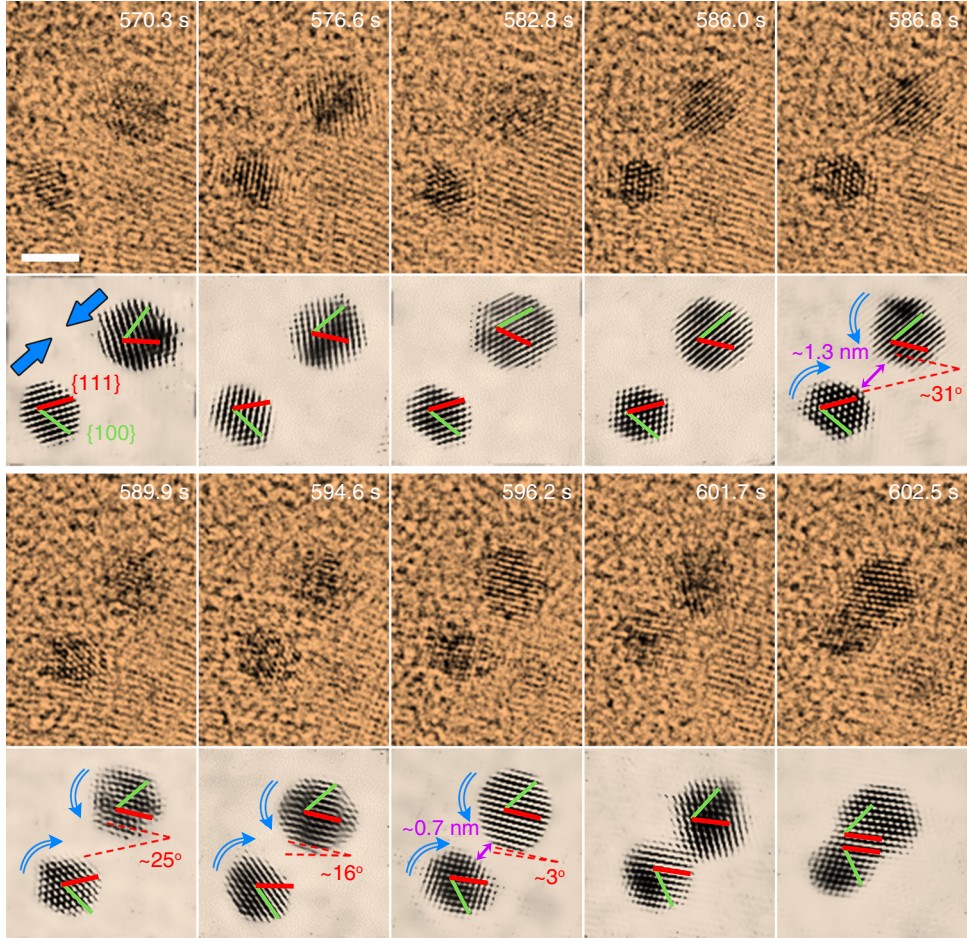

**Fig. 1** Imaging of OA at atomic level. Video sequences from Supplementary Movie 3 showing the OA process of small gold nanoparticles at {111} surface, evolving into a twin structure. The sequences below each false color TEM image are corresponding filtered images that highlight the evolution of particle orientations during OA, and also show the approaching of particles and the establishment of pre-alignment by rotation, after which the jump to contact occurs. In general, pre-alignment takes about tens of seconds while jump to contact is accomplished in less than 1 s. Red lines stand for {111} facets and green ones for {100} facets. Dashed lines depict the relative angle between the {111} facets of the two particles. The direction of movement of the particles (approaching and rotation) is denoted by blue arrows. Scale bar, 2 nm

there must exist a short-range mechanism which compels the particle pair to rotate directionally, resulting in an alignment of their {111} facets.

Generally, nanoscale interactions guiding the assembling of nanomaterials are described by van der Waals, electrostatic, magnetic and steric-hydration forces[34,35]. In our experiment, the Debye screen length can be estimated to be $\kappa^{-1} = 0.12$ nm (Supplementary Note 2), which means that the charged related electrostatic force between the gold particles can be ignored at separation distance larger than 0.12 nm. The dipolar interaction is also evaluated and its influence is ruled out in our case (Supplementary Fig. 13, Supplementary Note 2). Besides, the non-magnetic nature of the gold nanoparticles excludes the presence of a magnetic force. Therefore, for pairwise particles at $D > 0.12$ nm, the interaction potential can be expressed as follows[35,36]:

$$U(D) = W_0 e^{-\frac{D}{\lambda}} + \left\{ -\frac{A}{6} \left( \frac{2R^2}{(4R + D)D} + \frac{2R^2}{(2R + D)^2} + \ln \frac{(4R + D)D}{(2R + D)^2} \right) \right\} \quad (1)$$

where the first term is the steric-hydration repulsive potential due to the presence of ligands and the second term represents the van

der Waals attractive potential. We fit the potential by applying the distribution of pairwise separation distance for all the particle pairs (Supplementary Fig. 10, Supplementary Note 2)[37], as shown in Fig. 3a,b. The thickness of the citrate ligand layer adsorbed on the surface of the nanoparticles is thus calculated to be $L_{citrate} = \pi\lambda$ (0.21 nm) = 0.66 nm[35], which is in excellent agreement with the calculated thickness value for a single citrate layer on a gold surface[38]. As we can infer from Fig. 2, in stage I, coincidentally, all the separation distance are larger than 1.3 nm, which is the length of about $2L_{citrate}$. This suggests that at this stage, the citrate ligands on the two individual particle surfaces are not yet close enough to touch each other, which explains the reason why at this stage the particles rotate freely, resulting in random changes in the angle between their {111} facets. Nevertheless, as the long-range attractive force drags the two particles closer, the process evolves into stage II (separation distance smaller than $2L_{citrate}$), where their ligands inevitably begin to overlap. At this stage, the ligands act as a bridge by which the two particles become a single entity in that, their behaviors are no longer independent. On one hand, the particles continue to interact with each other by the combined effect of van der Walls and steric-hydration forces. On the other hand, some other mechanism, associated with the ligands, drives the particles to rotate gradually and directionally

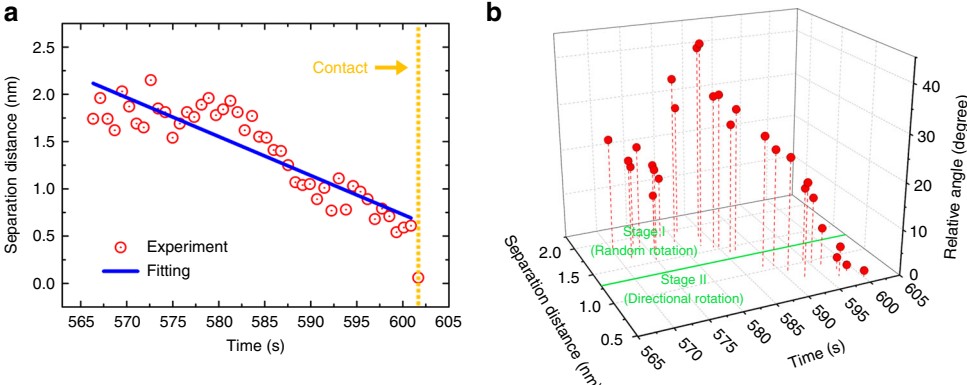

**Fig. 2** Dynamics of pairwise particles during OA. **a** Change in surface separation distance of a particle pair with time. The orange dashed line at 601.7 s denotes the occurrence of jump to contact. **b** The relative angle between the {111} facets of the particle pair vs. the separation distance and time. The green line at $D = 1.30$ nm is the separation point between stage I and stage II as the particle pair approaches each other

until their {111} facets are oriented parallel to each other. Eventually, at a separation distance of about 0.7 nm ($\sim L_{citrate}$), attachment occurs by a jump to contact of the aligned particles. The statistics of separation distance (Fig. 3c) indicate that all the particle pairs have a similar jump to contact distance of $L_{citrate}$, which implys that the particle surface contact results inevitably through the sudden explusion of their surface ligands. The general scheme of the detailed OA process is demonstrated in Fig. 3d.

**The role of binding energies confirmed by theoretical calculations.** We tracked OA events in more than 20 pairs of particles, and interestingly, we found that all these pairs preferentially contacted at {111} rather than at {100} or {110} surfaces (Supplementary Fig. 11). To uncover the ligand related mechanism of this observed preference for the spontaneous crystal facet selection, we calculated the surface energies, binding energies as well as ligand binding configurations of these surfaces using density function theory (DFT). {110} facets are omitted in our calculation because, unlike {111} and {100} facets, {110} facets of gold nanostructures have been widely recognized as unstable surfaces. On {110} surfaces, atom reconstruction such as the formation of zigzag atomic steps built with {111} or {100} sub-facets is a commonly observed phenomenon[39,40]. In other words, real {110} surfaces are not as flat as predicted by theory, and the presence of atomic steps could hinder a perfect attachment. Besides, for a face-centered cubic (fcc) structure, the {110} facets that have larger surface free energy have smaller surface areas than the other two facets according to Wulff construction. This is especially true in our case, where surface atoms occupy a relatively large proportion of the total number of atoms (particle size down to 2−3 nm), and therefore, nearly no clear imaging of {110} surfaces can be performed (Supplementary Fig. 5). In view of these drawbacks, the {110} surfaces are not ideal contact locations for OA and the competition is mainly between {111} and {100} facets. As summarized in Table 1, the un-passivated surface energies for {111} and {100} facets are respectively, 3.53 and 5.17 eV nm$^{-2}$. The ratio of surface areas between an individual {111} and {100} facet is estimated to be from 1:1.2 to 1:0.9, based on the highly symmetrical truncated octahedron shape of small particles (Supplementary Fig. 5). Hence in the absence of ligands, there is a lower probability of OA at the {111} surfaces by reason of the lower energy release during the elimination of surfaces.

However, since our nanoparticles are closely wrapped with citrate ligands, the contribution due to un-passivated surface energy is no longer important for the OA process to occur.

Instead, the ligand binding energy has to be taken into account, since the final jump to contact of the two surfaces will lead to the breaking of bindings between citrates and gold atoms and the followed expulsion of the ligands. It has been shown that the sodium citrate monolayer adsorbed on the gold nanoparticle surface is composed of di-hydrogen anions ($H_2Citrate^-$), and their central carboxylate groups serve to functionalize the surface atoms[38]. Accordingly, we have considered different possible ligand binding configurations (top, bridge and hollow) as initial states for simulation (Supplementary Note 3). Figure 4 shows the configurations of a citrate on {111} surface, from which it can been seen that after relaxation, the bridge and hollow configurations eventually transform to the top one, where the oxygen atoms of the central carboxylate group are located perpendicular to the top surface of gold atoms, suggesting that this is the most energetically stable state among the different structures. Configurations on {100} surfaces resemble those on {111} surfaces (Supplementary Fig. 12). In addition, in this top configuration, the thickness of the monolayer of citrate ligands is about 0.62 nm for both facets, which agrees perfectly with the experimental value of 0.66 nm. Thus, based on this configuration, the binding energies of a single citrate group is calculated to be 2.37 and 2.82 eV for {111} and {100} surfaces, respectively (Table 1). Apparently, under the pressure of two approaching particles, the weak binding ability (low binding energy) for {111} surfaces causes the adsorbed citrate ligands to be easily desorbed from the corresponding surfaces to make room for the particles to further approach each other, whereas the strong binding ability (high binding energy) for the {100} surfaces hinders this process. Therefore, we conclude that the parameter that plays a key role in controlling self-orientation during OA is the binding energy of the ligands at the different crystal facets.

**Discussion**

Despite the fact that several kinds of driving force for OA may exit, owing mainly to the great diversity of solution systems that have been investigated, the common factor is the following: in a great majority of chemical solutions for nano-scale synthesis, the surfaces of nanomaterials are invariably functionalized with organic ligands. Numerous studies have reported that surface ligands not only serve as good stabilizers to block the coalescence of nanoparticles, but sometimes these ligands also have an opposite, often positive effect on morphology control by facilitating particle coalescence at specific facets[4,25,41,42]. However, until now, all the conclusions on ligand induced OA are logical speculations based on ex-situ TEM characterization results. The

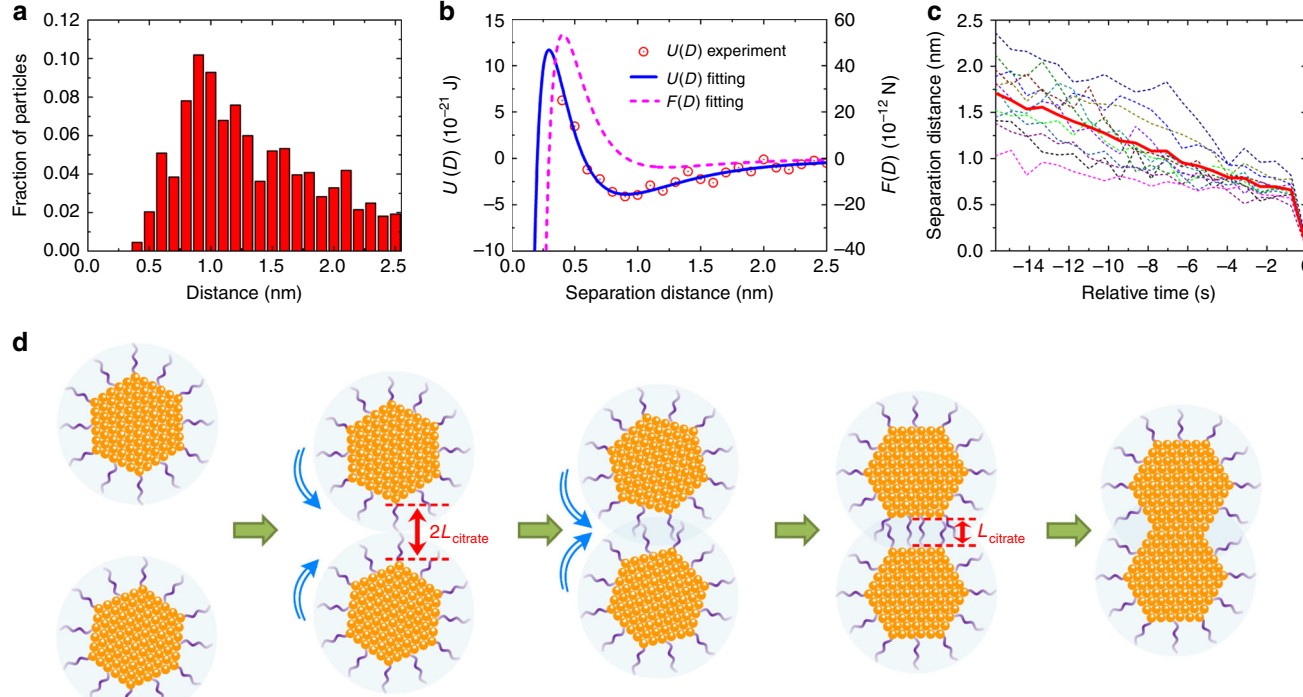

**Fig. 3** Interactions between pairwise particles. **a** A combined statistical distribution of all surface separation distance during the process of approaching in the observed OA events by 21 particle pairs. **b** Interaction potential of particle pairs as a function of their separation distance. The values for the potential (red dots) are extracted from statistics in (**a**) using Boltzmann distribution: $P(D) = C \exp(-U(D)/k_B T)$, where $U(D)$ is the combined potential energy resulting from the steric-hydration repulsive force and the van der Waals attractive force. The fitted potential and the corresponding force ($F(D) = -dU(D)/dD$) are shown by solid blue and dashed magenta curves, respectively. **c** Separation distance between 12 particle pairs during their approaching vs. relative time (while assuming that contact occurs at 0 s). Solid line is the average distance for these particle pairs. **d** Schematic illumination of the whole OA process

effect of ligands on OA remained unproven and the mechanism by which they drive OA has not been identified. Our work on in-situ TEM observations, undoubtedly offers a direct atomic level evidence that capping ligands do control the occurrence of OA at particular crystal facets, and this arises from their different binding abilities when ligands adsorb on different surfaces. It is also noteworthy that although the observed pre-alignment and jump to contact behaviors show some similarity with previous findings[32,33], deep analysis based on our quantitatively statistical data demonstrates a different mechanism which should be distinguished from those literatures, as well as more details of OA trajectories that have never been discovered before (Supplementary Note 4, Supplementary Fig. 14).

Besides, our proposed mechanism is not limited to simple cubic crystals like Au, Ag, and Pt. It also can extend to those compound growth systems, where the type and arrangement of atoms may fully differ at different surfaces of nanocrystals, which makes some other factors, such as dipole–dipole interaction[8,43], surface hydrolysis effect[42], and selective adsorption property[44] interfere the function of capping ligands on OA. However, even though these factors yield more complicated environments, extensive studies have declared the crucial role of ligands in control OA growth. X-ray diffraction (XRD) and ex-situ TEM indicate the structural distinctions between mercaptoethanol-caped and ligand-free ZnS through OA growth[45], as well as its ligand related two-stage coarsening mechanism that is dominant by specific OA[23]. Other studies of $TiO_2$ have demonstrated that adsorbed organic ligands can modify the OA process in way of steering the arrangement at selective crystal surfaces[46–48]. The experimental data for designing for PbSe nanostructures suggest that OA can selectively occur on either {100}, {110}, and {111}

facets to form various single crystalline morphologies, relying on the ligand molecules used in fabrication[25]. Moreover, increasing nanomaterials as diverse as CdSe[3], CuO[49], MnO[50], $PbWO_4$[51], $CaCO_3$[52], etc. have been designed by OA process with the assistance of organic ligands. All these experimental results imply the importance of surface ligands during OA. Last but not least, no matter what kind of materials, ligands definitely bond to the surface atoms of nanocrystals, so that desorption of ligands has to be considered as their surfaces come into contact. Therefore, our mechanism reported here provide a universal conceptual framework for understanding the OA process in depth, which should be applicable to a variety of ligand-mediated solution and materials including oxides, sulfides and semiconductors.

**Table 1 DFT calculations to determine the surface and binding energies for {111} and {100} surfaces**

| | Surface energy (eV nm⁻²) | Binding energy (eV per molecule) | | |
|---|---|---|---|---|
| | | Surface 1ᵃ | Surface 2 | Surface 3 |
| {111} | 3.53 | 2.36 | 2.40 | 2.37 |
| {100} | 5.17 | 2.77 | 2.83 | 2.87 |

Surfaces 1 to 3 have approximate values of binding energies, so we have used the average of the two values in the text, i.e., 2.37 eV for {111} surfaces and 2.82 eV for {100} surfaces
ᵃSurfaces 1 to 3 correspond to the three configurations shown in Fig. 4

## Methods
**Preparation of liquid cell samples.** Liquid solution containing gold nanoparticles with sizes of 10–20 nm were prepared by a simple redox reaction. 34 mM sodium

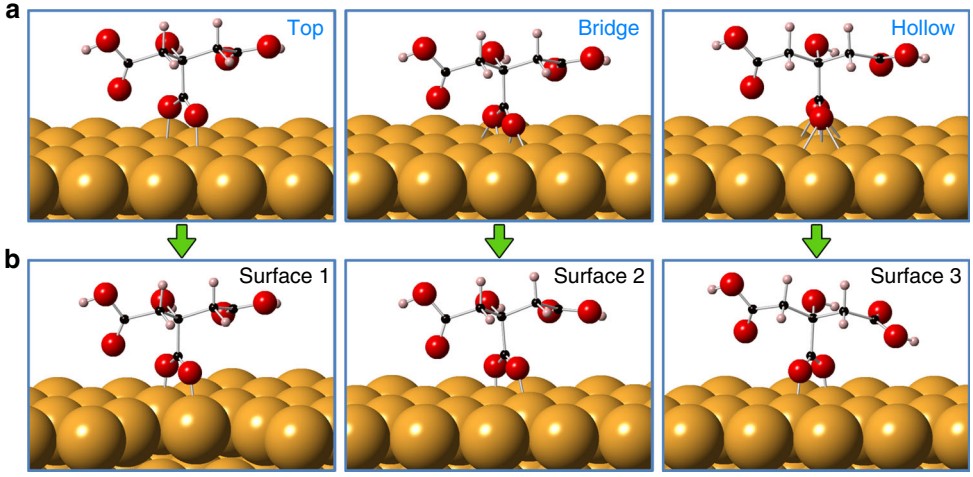

**Fig. 4** Configurations of citrate ligands adsorbed on gold {111} surfaces. **a** Possible configurations: Top, bridge and hollow, depending on the binding position of oxygen atoms, are applied as the initial states. **b** After relaxation, citrate in both bridge and hollow configurations change their location to the top of the gold atoms, indicating that the top configuration is the most stable one. Atoms are noted with colors (yellow: gold; red: oxygen; black: carbon; pink: hydrogen)

citrate aqueous solution was mixed with 0.24 mM $HAuCl_4$ aqueous solution until the color becoming soft pink. A droplet (2–3 µL) of the as-synthesized solution was sandwiched by two carbon film-covered faces of TEM grids to form a thin liquid layer between them. Then, the liquid cell was left to naturally dry under ambient atmosphere overnight. The van der Waals force of amorphous carbon film could lead to a small amount of solution was sealed in some pockets. Finally, the sample was transferred into a pre-vacuum system for 2 h to avoid the leakage.

**In-situ production of small nanoparticles and observation.** The in-situ observation was carried out using FEI Tian 80–300 TEM with Cs-corrector operated at 300 keV. Firstly, gold nanoparticles were exposed to electron beams at current density of $4$–$5 \times 10^5$ e $nm^{-2}$ $s^{-1}$ for tens of seconds. Then, due to the effect of irradiation, gold nanoparticles dissolved in solution and at the same time smaller nanoparticles with sizes around 2 nm were re-produced. After the dissolution process suspended or finished, the OA process of these generated small particles was observed and recorded.

**Data availability**. The authors declare that all the data supporting the findings of this study are available within the article and its Supplementary Information files. Other relevant data are available from the authors on reasonable request.

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

## Acknowledgements

This work was supported by the Projects of International Cooperation and Exchanges NSFC under grant Nos. 51420105003, the major program of National Science Foundation of China under grant Nos. 11327901, the National Natural Science Funds for Distinguished Young Scholar under grant Nos. 11525415, and the National Natural Science Foundation of China under grant Nos. 61274114, 11504046, 11674052.

## Author contributions

C.Z., H.Z., and L.S. conceived and designed the experiments. T.Z. and F.S. prepared the samples. C.Z. and W.W. performed the in-situ TEM imaging. S.L., Y.Z., E.S., and J.L. performed the theoretical calculations. C.Z., K.Y., S.L., C.H., and Y.Z. carried out the data analysis. C.Z., S.L., Y.S. H.Z., and L.S. co-wrote the paper. All authors discussed the results and revised the manuscript.

## Additional information

**Competing interests:** The authors declare no competing financial interests.

