## [Peer Review File · Nature Communications]

Reviewers' comments:

Reviewer #1 (Remarks to the Author):

Understanding the nucleation and growth of crystals from solution is essential to control the morphology and structure of nanocrystals. In this manuscript, the authors investigated the oriented attachment (OA) of gold nanoparticle using liquid cell transmission electron microscopy (TEM) at atomic level, and suggested the important role of surface ligands in controlling the OA process. I would recommend this manuscript to be accepted. However, the following concerns need to be addressed.

In this work, the authors fabricated liquid cell using two carbon films and showed the existence of solution through an image of liquid pocket, but this evidence is not convincing. Actually, according to the preparation procedure of liquid cell samples in the manuscript, it is hard to believe that the sealing of liquid cell is good enough to avoid the evaporation of solution just simply putting two carbon films together. If ever there exists liquid in the cell, it is likely that the solution evaporated completely after overnight drying under ambient atmosphere and pre-vacuuming for 2 hours (the authors indeed indicated the continuous evaporation of solvent during the cell fabrication). Is it possible that the in-situ observation was probably performed in vacuum instead of in solution? The metal nanoparticles (e.g. Au, Ag) show migration and coalescence in vacuum under electron beam irradiation. (See references: Chem. Commun., 2013, 49, 11479; Sci. Rep. 2016, 6, 21498)

From Line 100 to Line 102, the authors stated that large gold nanoparticle were consumed while smaller ones appeared. From the viewpoint of thermodynamics, smaller particles have higher specific surface area and hence show inferior stability compared to large one. Can the authors give detailed explanations regarding this matter?

After mixing sodium citrate and HAuCl_4 aqueous solutions, the resulting solution consists of $\text{C}_6\text{H}_5\text{O}_7$, HCl and NaCl , so is it still accurate to calculate the Debye length only based concentration of saturated NaCl solution? Actually, the estimated Debye length (0.12 nm) based on saturated NaCl solution is smaller than the thickness of the citrate ligand layer adsorbed on the surface of the nanoparticles (0.66 nm). "electrostatic force between the gold particles can be ignored at separation distances larger than 0.12 nm" Does the electrostatic force affect the attachment? The authors have shown that the separation is more or less around 0.5 nm right before attachment and showed the directional rotation within 1.25 nm. Do they all finish alignment beyond 0.5nm separation for the 20 pair particles? Do they all show directional rotation within 1.25 nm? Author may need to do some control experiments to demonstrate the role of ligands in oriented attachment.

There are some mistakes in this manuscript. For example, Line 153, the inequality should be $D < 1.3$ nm; Line 48 in supplementary information, the equation should be $0.304/\sqrt{[\text{NaCl}]}$. The authors should carefully revised the manuscript.

Reviewer #2 (Remarks to the Author):

The presenting work reported in-situ investigations on OA growth of gold nanoparticles, and tried to discuss the driving force of OA. It provides some interesting insights on revealing the OA phenomena, but scarcely shed new lights on its mechanism and driving force. Several intrinsic shortcomings are shown as follows:

1. The first part concerned on the relationship between the OA-growth speed and the distance (or contact angle) of particles. It should be noted that, previously, several studies on OA growth via liquid Cell TEM techniques have already revealed that, "the drift velocity increase greatly at very close approach, indicating the strong attractive forces at short range (Zheng et al, Science 2012, 336, 1011)". In addition, the rotation and alignment trajectories of two attaching particles have also been captured (Yoreo et al. in Science, 2012, 336, 1014). These published papers would make the innovation of presenting work considerably weakened, since it just has similar data and conclusion without in-depth (new) analysis/discussion.

2. Questions are also on the preferential orientation of OA among gold nanoparticles. As described in Line 129-131, "A monocrystalline structure can be formed in the same way if both the {111} and {100} facets are aligned before contact". It demonstrates the OA can happen on both the {111} and {100} facets. However, in line 208-209, "We tracked OA events in more than 20 pairs of particles, and interestingly, we found that all these pairs preferentially contacted at {111} rather than at {100} or {110} facets." Such two conclusions are contradictory with each other. By the way, as shown in Fig. S7, some of figures has very low resolution, from which it is hard to identify the preferential orientation facets. Therefore, I worried about if the OA preferential happened on {111}. Generally, in many experiments, the OA frequently occurred on different facets of particles. This makes the following discussing of the driving force of OA (solely on {111}) questionable.

3. The big intrinsic problem is that it did not provide a clear and solid evidence for the driving force of OA. It ruled out the contribution of specific surface energy, and then assigned it to the desorption of surfactant. Making a conclusion in this way is illogical. Other comments include: i) The surface area of each facet was not taken into account when discussing the contribution of surface energy. ii) The surfactant and pH value might change the surface charge of gold nanoparticles, which affects the interaction force of adjacent particles and their OA behavior. Did this factor be taken into account? And what happened once the pH and ligands changed? iii) "In addition, the thickness of the monolayer of citrate ligands is about 0.62 nm for both facets, which agrees perfectly with the experimental value of 0.65 nm." Contradictorily, the separation distance shown in Fig.3a has a distribution range. Moreover, no clear evidence demonstrates the direct relationship between these two values. In previous work (Zheng et al, Science 2012, 336, 1011), the fast OA growth also occurred in short distance (no more than 1 nm). iv) After reading it carefully for several times, I still do not know what is the driving force of OA among gold nanoparticles.

4. In my opinion, the author had better add some details about the progresses on the driving force (or mechanism) of OA in recent literatures in Introduction part.

Based on these shortcomings, it seems the presenting work in this version is not suitable for publication in Nature Communications.

Reviewer #3 (Remarks to the Author):

Based on in-situ TEM, the work reports the oriented motion evolution of attaching Au nanoparticles at separation distances close to twice the layer thickness of adsorbed ligands. The oriented attachment growth is well illustrated. I have three major concerns which require major revisions of the work prior to publication:

1. The evaluation of ligand thickness isn't clearly presented. How did the authors obtain the λ value in

their analysis?

2. The authors didn't take dipolar interaction into consideration in their calculations. The steric hinderance of ligands can vary substantially as attaching crystals get close. These factors should be assessed very carefully.

3. Au is a simple cubic crystal. How does the reported method contribute to the understanding on the oriented attachment of more complicated crystals, including oxides and other compound crystals?

Response to the reviewers' comments

Reviewer #1 (Remarks to the Author):

Understanding the nucleation and growth of crystals from solution is essential to control the morphology and structure of nanocrystals. In this manuscript, the authors investigated the oriented attachment (OA) of gold nanoparticle using liquid cell transmission electron microscopy (TEM) at atomic level, and suggested the important role of surface ligands in controlling the OA process. I would recommend this manuscript to be accepted. However, the following concerns need to be addressed.

We thank this referee for her/his positive appreciation of our work.

Comment 1: In this work, the authors fabricated liquid cell using two carbon films and showed the existence of solution through an image of liquid pocket, but this evidence is not convincing. Actually, according to the preparation procedure of liquid cell samples in the manuscript, it is hard to believe that the sealing of liquid cell is good enough to avoid the evaporation of solution just simply putting two carbon films together. If ever there exists liquid in the cell, it is likely that the solution evaporated completely after overnight drying under ambient atmosphere and pre-vacuuming for 2 hours (the authors indeed indicated the continuous evaporation of solvent during the cell fabrication). Is it possible that the in-situ observation was probably performed in vacuum instead of in solution? The metal nanoparticles (e.g. Au, Ag) show migration and coalescence in vacuum under electron beam irradiation. (See references: Chem. Commun., 2013, 49, 11479; Sci. Rep. 2016, 6, 21498)

Authors' Reply:

In the liquid cell fabrication process, most of the solution evaporates after overnight drying, while a small amount is well encapsulated by amorphous carbon films to form liquid pockets, as shown in Figure R1a. It is found that solution is still available inside some liquid pockets after 3 days, indicating the great sealability of carbon films.

Besides, we confirm the observed OA occurs in solution instead of in vacuum on account of

the following two reasons. First, the manner of nanoparticle motion in vacuum is different from that in solution. In vacuum, nanoparticles are supported by carbon films and they always move in the manner of slow translation (*Chem. Commun.*, 2013, 49, 11479; *Sci. Rep.*, 2016, 6, 21498). In comparison, particles can undergo free translation and rotation in solution. Figure R1b is the TEM sequences to show the motion of a “L” shape nanostructure in the liquid pocket. It does not lay on the carbon films, but keeps rolling freely (not only along the viewing axis but also off axis) because it is suspended in solution. Second, only in the liquid environment, the dissolution and precipitation can take place to produce small nanoparticles (Supplementary Information, page 2, line 50-62). We have tried to irradiate the large particles in vacuum using the same electron dose, but no dissolution and precipitation phenomena have been observed. Another video example about the motion, dissolution and precipitation of gold particles has been added (Supplementary Movie 2) to support the fact that all the results are obtained inside liquid environment.

Figure R1. **a**, TEM image of a liquid pocket (some bright fringes inside the liquid pocket come from the diffraction contrast of precipitated NaCl crystal). **b**, Free rolling of “L” shape nanostructure.

Comment 2: From Line 100 to Line 102, the authors stated that large gold nanoparticle were consumed while smaller ones appeared. From the viewpoint of thermodynamics, smaller particles have higher specific surface area and hence show inferior stability compared to large one. Can the authors give detailed explanations regarding this matter?

Authors’ Reply:

Thanks for the pertinent suggestion. Indeed in an ordinary equilibrium solution system, this

anti-Ostwald ripening evolution seems unlikely to happen. However for our case, the electron beam plays an important role in the dissolution of large gold particles. The incident high energy electrons can react with water molecules to produce some oxidation agents such as H_2O_2 , OH , O and O_2 . These agents then significantly facilitate the dissolution of gold atoms in the presence of H^+ and Cl^- which are the residual of the reaction solution. Eventually, with continuous dissolution, the concentration of AuCl_4^- increases, leading to the followed precipitation of small particles. We have added a detailed discussion about the dissolution-precipitation phenomena in Supplementary Section 1 (Supplementary Information, page 2, line 37-62)

Comment 3: After mixing sodium citrate and HAuCl_4 aqueous solutions, the resulting solution consists of $\text{C}_6\text{H}_5\text{O}_7$, HCl and NaCl , so is it still accurate to calculate the Debye length only based concentration of saturated NaCl solution? Actually, the estimated Debye length (0.12 nm) based on saturated NaCl solution is smaller than the thickness of the citrate ligand layer adsorbed on the surface of the nanoparticles (0.66 nm). “electrostatic force between the gold particles can be ignored at separation distances larger than 0.12 nm” Does the electrostatic force affect the attachment? The authors have shown that the separation is more or less around 0.5 nm right before attachment and showed the directional rotation within 1.25 nm. Do they all finish alignment beyond 0.5nm separation for the 20 pair particles? Do they all show directional rotation within 1.25 nm? Author may need to do some control experiments to demonstrate the role of ligands in oriented attachment.

Authors’ Reply:

We thank the reviewer for this suggestion. Our solution should contains Na^+ , H^+ , Cl^- , $\text{C}_6\text{H}_5\text{O}_7^{3-}$ and AuCl_4^- . We cannot determine the concentrations for all these ions because of the evaporation of solvent during liquid cell fabrication, which makes it difficult to evaluate the exact Debye length. But if all these ions are taken into account, the Debye length is even smaller than 0.12 nm (see detailed analysis in Supplementary Information, page 3, line 87-99). Therefore, the electrostatic force do not affect the OA due to the electrostatic screen when the separation of the particles is larger than 0.12 nm.

We have carried out some control experiments to verify our results. All the particle pairs get well aligned before contact at about 0.7 nm, despite that for some pairs the relative angle of their {111} facets is not perfectly 0° but always less than 5°. As for the directional rotation, it indeed occurs around 1.3 nm for most cases. Figure R2 and R3 are two other examples to show the distance and angle change of particle pairs.

Figure R2. In this case, the angle changes randomly when $D > 1.2$ nm. Then as the particle pair gets closer ($D < 1.2$ nm), the directional rotation begins to make the angle gradually decrease to approximate 0°.

Figure R3. In this case, the angle does not change much as $D > 1.3$ nm. But when the particle pair approaches to a smaller distance ($D < 1.3$ nm), it quickly decreases to nearly 0° .

Comment 4: There are some mistakes in this manuscript. For example, Line 153, the inequality should be $D < 1.3$ nm; Line 48 in supplementary information, the equation should be $0.304/\sqrt{[\text{NaCl}]}$. The authors should carefully revised the manuscript.

Authors' Reply:

We have carefully corrected these mistakes in the manuscript and the revised manuscript has been significantly improved.

The word “interact” is modified to “overlap” (page 2, line 55, page 6, line 200). The sentence “in the range 10-20 nm” is changed to “in the range of 10-20 nm” (Page 3, line 102). “(111)” is changed by “{111}” (page 4, line 148; page 7, line 213). The sentence “this critical distance 1.3 nm is independent” is modified to “this critical distance is always around 1.3 nm and independent” (page 5, line 164). The inequality “ $D < 0.13$ nm” is corrected to be “ $D < 1.3$ nm” (page 5, line 165). The word “interaction” is replaced by “mechanism” (page 5, line 166). The word “facets” is modified to “surfaces” (page 7, line 220, 223, 234, 240, 251, 255, 259 and 261; page 8, line 263, 268 and 270). The word “energies” is corrected to “energy” (page 7, line 231). The word “citrate” is corrected to “citrates” (page 7, line 245). “0.65 nm” is corrected to be “0.66 nm” (page 7, line 257). The word “negative” is corrected to be “positive” (page 8, line 278)

The equation “ $3.04/\sqrt{[\text{NaCl}]}$ ” is corrected to be “ $0.304/\sqrt{[\text{NaCl}]}$ ” (Supplementary Information,

page 3, line 97).

Reviewer #2 (Remarks to the Author):

The presenting work reported in-situ investigations on OA growth of gold nanoparticles, and tried to discuss the driving force of OA. It provides some interesting insights on revealing the OA phenomena, but scarcely shed new lights on its mechanism and driving force. Several intrinsic shortcomings are shown as follows:

Comment 1: The first part concerned on the relationship between the OA-growth speed and the distance (or contact angle) of particles. It should be noted that, previously, several studies on OA growth via liquid Cell TEM techniques have already revealed that, “the drift velocity increase greatly at very close approach, indicating the strong attractive forces at short range (Zheng et al, Science 2012, 336, 1011)”. In addition, the rotation and alignment trajectories of two attaching particles have also been captured (Yoreo et al. in Science, 2012, 336, 1014). These published papers would make the innovation of presenting work considerably weakened, since it just has similar data and conclusion without in-depth (new) analysis/discussion.

Authors' Reply:

We respectfully disagree the assertion about the innovation as thanks to the reviewer's comments.

In our previous work (Zheng et al., *Science*, 2012, 336, 1011), it is found that polycrystalline nano-chains can be built by imperfect attachment of nanoparticles when the dipolar interaction compels the particles to arrange end-by-end. In that work, the drift velocity increases greatly at very close distance because the attractive dipolar interaction ($U \sim 1/r^2$) increases dramatically as the particles get closer. This increase of dipolar interaction also leads to an apparent acceleration process when the distance decreases to 3 nm for particle-particle pair and 6 nm for particle-chain pair (Figure 4A, *Science* 2012, 336, 1011). However, the situation is different in this work. The attractive force does not change much as

the particles get closer, and then even becomes repulsive force when the distance is smaller than about 1 nm (please see the dashed magenta curves in Figure 3b, this manuscript). This is why the drift velocity slightly decreases as they closer (Figure R4). In addition, the abrupt increase of velocity before the final contact (Figure R4) well indicates that the expulsion of surface ligands leads to the vanishing of repulsive force and then quick contact. Therefore the jump to contact in this work should be distinguished from the increase of drift velocity in our previous work, similar behavior but different mechanism.

Figure R4. Relative drift velocity of a particle pair when they approach each other

In another previous work (Yoreo et al., *Science*, 2012, 336, 1014), the pathways of OA in solution has been in-situ observed. Although the authors have imaged the successive pre-alignment, jump to contact and interface elimination process, they have not performed quantitatively statistical analysis about the detailed evolvement laws of particle's movement and rotation before contact. In comparison, here in our work, the small size of gold particles allows us to study the movement and rotation behavior more easily, and a large number of statistical data also allow us to investigate the nature of driving force. More details about OA have been discovered in our work.

Most importantly, numerous literature about nano-synthesis have noticed the crucial role of surface ligands in OA, whereas there is still no direct evidence to elucidate this mechanism. Our results have, for the first time, revealed how the surface ligands control OA and provided a new mechanism of the binding energy related crystal facet selection. These new findings

were enabled by the liquid cell TEM with significant improved spatial resolution as compared to the previous reports. Owing to these significant findings in our work that were not achieved in the previous publications, we can conclude that undoubtedly our work possesses bold innovation and it has provided novel experimental observation that was previously not reachable as well as depth analysis/discussion. It may attract readers in different fields.

Comment 2: Questions are also on the preferential orientation of OA among gold nanoparticles. As described in Line 129-131, “A monocrystalline structure can be formed in the same way if both the {111} and {100} facets are aligned before contact”. It demonstrates the OA can happen on both the {111} and {100} facets. However, in line 208-209, “We tracked OA events in more than 20 pairs of particles, and interestingly, we found that all these pairs preferentially contacted at {111} rather than at {100} or {110} facets.” Such two conclusions are contradictory with each other. By the way, as shown in Fig. S7, some of figures has very low resolution, from which it is hard to identify the preferential orientation facets. Therefore, I worried about if the OA preferential happened on {111}. Generally, in many experiments, the OA frequently occurred on different facets of particles. This makes the following discussing of the driving force of OA (solely on {111}) questionable.

Authors’ Reply:

We thank the reviewer for raising these questions. The sentence “A monocrystalline ... if both the {111} and {100} facets are aligned before contact” is not intended to demonstrate the OA can happen on both {111} and {100} facets, but just tells how a monocrystalline structure is formed. We agree with the reviewer that this expression is ambiguous, so we have modified the manuscript to make a clear description (page 4, line 137-138, 140-142). Actually, as we describe in the sentence “We tracked ... rather than at {100} or {110} facets”, the contact always occurs at {111} instead of {100} surfaces because {111} facets are confirmed to be the joint facets at neck locations (Fig. S7). Hence, the prerequisite for contact is the alignment of {111} surfaces. Then there are two types of contact (see the schematic illustration in Figure R5): First, their {100} facets are parallel and a monocrystalline structure

is formed; Second, their {100} facets are not parallel and a twin structure is obtained. For both types, they contact at {111} surfaces.

Figure R5. Schematic diagram about the formation of monocrystalline and twin structures.

In addition, we have modified Supplementary Figure 8 (Supplementary Information, page 12, line 336-341). We add more examples to support our conclusion that all the observed OA occurs at {111} surfaces. Although some figures have low resolution, the corresponding FFT images of the neck locations make us confirm the {111} orientation. For all cases, the neck is formed by joint {111} facets. Indeed, in many experiments, the OA frequently occurred on different facets of particles. But at same time, it can controlled by changing experimental conditions. For instance, usage of different organic ligands can manage the occurrence OA of PbSe on different surfaces (Cho et al., *J. Am. Chem. Soc.*, 2005, 127, 7140). Even of Au nanoparticles, some groups have also noticed that OA only occur at their {111} facets (Halder et al., *Adv. Mater.*, 2007, 19, 1854). Therefore, the preference of OA at {111} facets in our case just implies the effect of surface ligands.

Comment 3: The big intrinsic problem is that it did not provide a clear and solid evidence for the driving force of OA. It ruled out the contribution of specific surface energy, and then assigned it to the desorption of surfactant. Making a conclusion in this way is illogical. Other comments include: i) The surface area of each facet was not taken into account when discussing the contribution of surface energy. ii) The surfactant and pH value might change the surface charge of gold nanoparticles, which affects the interaction force of adjacent particles and their OA behavior. Did this factor be taken into account? And what happened once the pH and ligands changed? iii) “In addition, the thickness of the monolayer of citrate ligands is about 0.62 nm for both facets, which agrees perfectly with the experimental value of 0.65 nm.” Contradictorily, the separation distance shown in Fig.3a has a distribution range.

Moreover, no clear evidence demonstrates the direct relationship between these two values. In previous work (Zheng et al, *Science* 2012, 336, 1011), the fast OA growth also occurred in short distance (no more than 1 nm). iv) After reading it carefully for several times, I still do not know what is the driving force of OA among gold nanoparticles.

Authors' Reply:

We thank the reviewer for these comments. But we believe it is reasonable to take the desorption of ligands into account. Firstly, theoretical simulation shows that if nanoparticles are capped with ligands the passivated surface energy may significantly changes with the coverage density of ligands, which leads to either positive or negative difference of surface energy values between {111} and {100} facets (Bealing et al., *ACS Nano*, 2012, 6, 2118). This means in our case {111} facets may have larger passivated surface energy than {100} facets. But, more importantly, no matter which facet has larger passivated surface energy, the strong bonding ability of ligands at {100} surfaces will prevent the desorption of ligands and followed OA at this facet. Additionally, some other groups have also noticed the important role of bonding ability of ligands in OA events. Polleux et al. have claimed that the selective desorption of ligands can result in the formation of TiO₂ in [001] direction (*Adv. Mater.*, 2004, 16, 436). Zhang et al. show that the occurrence of OA should be related to the easily destroyed ligands (*J. Phys. Chem. B*, 2007, 111, 1449). Halder et al. also speculate the difference in amine/gold binding energy on different facets enables preferential removal of the ligands from one of the facets (*Adv. Mater.*, 2007, 19, 1854). Therefore, the bonding ability should be the decisive factor instead of surface energy.

i) We have added a discussion about the surface area of each facet when talking about the surface area (page 7, line 237-239) and modified the related figure (Supplementary Figure 4, page 10, line 288, 291, 304-309). The surface energy of individual {100} facet is still larger than that of individual {111} facet.

ii) We have added more details about the solution environment (Supplementary Information, page 2, line 38-50), and some discussion about the concentration of ions (Supplementary

Information, page 3, line 87-99). Certainly pH value and ligands may change the surface charge of gold nanoparticles, but even if surface charge changes, the Debye length is still estimated to be smaller than 0.12 nm for our multi-ionic system (Supplementary Information, section 2). It means the charge induced interaction force is screened unless the separation of particles is smaller than 0.12 nm; however all the approaching, rotation and pre-alignment are complete before jump to contact distance (~ 0.7 nm). That's why we ignore the charge interaction. Besides, citrates are reported to have similar adsorption configuration on {111}, {100} and {110} facets of gold nanoparticles with negative surface charges (Park et al., *J. Am. Chem. Soc.*, 2014, 136, 1907), and we have shown that the small gold nanoparticles have symmetrical morphology (Supplementary Figure 4). So the surface charge interaction at most has a repulsive effect between particles, but will not lead to their directional rotation. Based on these reasons, we believe the surface charges do not contribute much to OA in our case.

iii) Figure 3a plots the combined distribution of all separation distance for 21 particle pairs during their approaching process. It is not a distribution of the jump to contact distance for 21 particle pairs. Hence, this distribution actually reflects the counts of every possible separation distance as the particles approach. We have modified the figure caption to make a more clear description (page 5, line 169-170). On the other hand, it should be firstly noted that repulsive steric force arises from the soft feature of surface ligands. When the ligand outer segments of approaching particles begin to overlap, they create the steric force upon compression of the soft "brushes" between the surfaces (Israelachvili, J. N. *Intermolecular and surface forces: revised third edition*). This interaction obeys $U_{\text{steric}} \propto \exp(-\pi D/L)$, where D is the separation distance of particle surfaces and L is the thickness of ligand. Then by fitting Figure 3a (see details in Supplementary Information, page 4, line 101-124), the experimental value $L = 0.66$ nm is obtained. This means the thickness of the citrates on particle surface is 0.66 nm. In the theoretical calculation, the thickness is obtained to be 0.62 nm, which well indicates the validity of our theoretical model as compared with experimental results. One is experimental value, another one is theoretical value, and both stand for the thickness of citrate. Apparently, these two values are closely related.

It should be emphasized again that when the distance decrease the compression between ligands become dominant and thus the repulsive force increases (Figure 3b), which leads to the deceleration of approaching velocity (Figure R4). It is different from the previous work (Zheng et al., *Science*, 2012, 336, 1011) as discussed comment 1.

iv) According to our analysis, the driving force for OA of gold nanoparticles is a combination of physical interaction which is van der Waals force, and thermodynamics mechanism which is the less energy cost through the expulsion of weakly binded ligands. The former is responsible for approaching, and the latter steers rotation (alignment). In our paper, we mainly focus on the role of ligands. The whole process is elucidated as following: Under the attraction of van der Waals force, the particle pair keeps approaching each other. When their ligands begin to overlap, the behavior of these two particles is then related because their connected ligands make them become a single entity. At this moment, the particles directionally rotate and align their {111} rather than {100} surface due to the weaker binding ability of citrates on {111} surface. Then, the approaching particles expel the citrates between their {111} surface by overcoming its smaller binding energy. Eventually, the particles contact to coalesce into a single particle.

Comment 4: In my opinion, the author had better add some details about the progresses on the driving force (or mechanism) of OA in recent literatures in Introduction part.

Authors' Reply

We thank the reviewer for this suggestion. We have followed the reviewer's suggestion and modified the Introduction part to summarize the recent progress on the driving force of OA (page 2-3, line 72, 75-90, 93-98, 100).

Reviewer #3 (Remarks to the Author):

Based on in-situ TEM, the work reports the oriented motion evolution of attaching Au

nanoparticles at separation distances close to twice the layer thickness of adsorbed ligands. The oriented attachment growth is well illustrated. I have three major concerns which require major revisions of the work prior to publication:

We thank the referee for her/his positive comments and appreciation of our work.

Comment 1: The evaluation of ligand thickness isn't clearly presented. How did the authors obtain the λ value in their analysis?

Authors' Reply

We thank the reviewer for this comment. The λ value is obtained from the fitting of the distribution of separation distance, which satisfies the Boltzmann statistics. We have modified the analysis of interaction potential and added detailed derivation of the decay length λ as well as the thickness of citrate ligand (Supplementary Information, page 4, line 101-124).

Comment 2: The authors didn't take dipolar interaction into consideration in their calculations. The steric hinderance of ligands can vary substantially as attaching crystals get close. These factors should be assessed very carefully.

Authors' Reply

Thanks for these suggestions. We have added some discussion about the dipolar interaction (Supplementary Information, page 3, line 74-99). The contribution of dipolar interaction to OA process is ignored in our system based on two factors: First, our gold nanoparticles possess nearly no dipole moment; Second, the ions in our solution have a screening effect on the electrical dipolar interaction.

The steric-hydration force in our case originates from two citrate-covered surfaces approaching each other. Once the ligands begin to contact or overlap, it leads to a repulsive force due to the compression of ligands between the surfaces. Indeed, when the attaching crystals get close, this interaction significantly increases since it obeys a power law as described by the equation in the main text (Israelachvili, J. N. *Intermolecular and surface forces: revised third edition*). The resultant force then changes from attraction to repulsion

and the equilibrium point where steric force equals to Van der Waals force is at about $D=0.9$ nm (Figure 3b). In some cases, as shown in Figure R6, the separation distance of particle pair oscillates around the equilibrium point ($D=0.9$ nm). It suggests that when D is very small (for example 0.59 and 0.56 nm in the images below), the steric force is strong enough to prevent particle pairs from coalescence. However at the weak binding surfaces ($\{111\}$ facets), the ligands are always desorbed at about $D=0.7$ nm. This indicates that the steric hindrance has not increased to a high magnitude before the ligands detach from particle surfaces.

Figure R6. The approaching particle pair is stabilized by surface citrates, and show as oscillation behavior around its equilibrium position.

Comment 3: Au is a simple cubic crystal. How does the reported method contribute to the understanding on the oriented attachment of more complicated crystals, including oxides and other compound crystals?

Authors' Reply

We thank the reviewer for raising a question many readers might have regarding. Nowadays, oriented attachment dominated growth in the presence of surface ligands have been reported for a lot of nanomaterials including oxides, sulfides and semiconductors. It suggests the great importance of the ligands while the detailed mechanism is still unclear. We were able to capture significant statistical data of small nanoparticle attachment with high resolution in situ. This opens the door to elucidating the role of ligands in OA of nanoparticles that previously has not been able to. Besides, regardless of what materials and organic additives, desorption of ligands is an inevitable behavior to realize the surface contact of two particles, which makes the binding between ligands and surface atoms play a crucial role. Based on all

these facts, our proposed mechanism here should be also suitable for the interpretation of oriented attachment process of other more complicated crystals. We have added a detailed discussion about this at the last part of manuscript (page 8-9, line 286-307).

Summary of the changes:

(Changes in the revised manuscript are highlighted in blue color)

1. In revised manuscript, page 1, line 4-5, the authors “Wen Wang, Feng Shan and Tong Zhang” are added because they have contributions to sample preparation and in-situ characterization during supplementary control experiments and the revision of manuscript.
2. In revised manuscript, page 1, line 12-14, “Joint International Research Laboratory of Information Display and Visualization, School of Electronic Science and Engineering, Southeast University, Nanjing 210096, P. R.” is added.
3. In revised manuscript, page 2, line 55, “interact” is modified to “overlap”.
4. In revised manuscript, page 2, line 58-59, “facet(s)” is modified to “surface(s)”
5. In revised manuscript, page 2-3, line 72, 75-90, 93-98, 100. The Introduction part is carefully modified and the recent progress on the driving force of OA is summarized.
6. In revised manuscript, page 3, line 102. “in the range 10-20 nm” is changed to “in the range of 10-20 nm”.
7. In revised manuscript, page 3, line 112. “Supplementary Movie 1” is changed to “Supplementary Movie 1, 2 and Section 1”.
8. In revised manuscript, page 3, line 114, “Movie 1 and 2” is changed to “Movie 3”.
9. In revised manuscript, page 4, line 129, “Movie 2” is changed to “Movie 3”.
10. In revised manuscript, page 4, line 129, “facet” is changed to “surfaces”.
11. In revised manuscript, page 4, line 137-138, 140-142, “leaving the joint {111} facet as a twin interface” is modified to “The contact at their {111} surfaces leaves this crystal facet as a twin interface since the angle between their {100} facets is about 70°”; “A monocrystalline structure can be formed in the same way if both the {111} and {100}

facets are aligned before contact (Supplementary Fig. 1 and Movie 3)” is modified to “A monocrystalline structure can be formed in the same way if the particles contact at aligned {111} surfaces but their {100} facets are parallel with each other (Supplementary Fig. 5 and Movie 4)”.

12. In revised manuscript, page 4, line 148, “(111)” is changed to “{111}”.
13. In revised manuscript, page 5, line 164, “1.3 nm is” is changed to “is always around 1.3 nm and”.
14. In revised manuscript, page 5, line 165, “ $D < 0.13$ nm” is corrected to “ $D < 1.3$ nm”.
15. In revised manuscript, page 5, line 166, “interaction” is modified to “mechanism”.
16. In revised manuscript, page 5, line 169-170, “A statistical distribution of the surface separation distances during the process of approaching in all the observed OA events by 21 particle pairs” is changed to “A combined statistical distribution of all surface separation distance during the process of approaching in the observed OA events by 21 particle pairs”.
17. In revised manuscript, page 5, line 181, “supplementary text” is changed to “Supplementary Section 2”.
18. In revised manuscript, page 6, line 190, “supplementary text” is changed to “Supplementary Fig. 7 and Section 2”.
19. In revised manuscript, page 6, line 200, “interact” is modified to “overlap”.
20. In revised manuscript, page 7, line 213, “(111)” is changed to “{111}”.
21. In revised manuscript, page 7, line 220, 223, 234, 240, 251, 255, 259, 261, “facet(s)” is changed to “surface(s)”.
22. In revised manuscript, page 7, line 221, “Supplementary Fig. 7” is changed to “Supplementary Fig. 8”.
23. In revised manuscript, page 7, line 230, “energies” is changed to “energy”.
24. In revised manuscript, page 7, line 237, “ nm^{-2} ” is corrected to “ eV nm^{-2} ”.
25. In revised manuscript, page 7, line 237-239, we have added “The ratio of surface areas between an individual {111} and {100} facet is estimated to be from 1 : 1.2 to 1 : 0.9, based on the highly symmetrical truncated octahedron shape of small particles (Supplementary Fig. 4)” to take the surface area of each facet into account when talking about the surface area.
26. In revised manuscript, page 7, line 245, “citrate” is corrected to “citrates”.
27. In revised manuscript, page 7, line 250, “supplementary text” is changed to “Supplementary Section 3”.
28. In revised manuscript, page 7, line 255, “Supplementary Fig. 8” is changed to “Supplementary Fig. 9”.

29. In revised manuscript, page 7, line 257, “0.65 nm” is changed to “0.66 nm”.
30. In revised manuscript, page 8, line 263, 268, 270, “facet(s)” is changed to “surface(s)”.
31. In revised manuscript, page 8, line 273, “there is still much debate on the nature of the driving force in OA processes” is changed to “several kinds of driving force for OA may exist”.
32. In revised manuscript, page 8, line 278, “negative” is corrected to “positive”.
33. In revised manuscript, page 8-9, line 286-307, we have added a discussion about the contribution of our mechanism to the understanding on the oriented attachment of more complicated crystals.
34. In revised manuscript, page 11, line 431, “T. Z. and F. S. prepared the samples” and “W. W.” is added to state their contributions.
35. In revised supplementary information, page 1, line 4-5, “Wen Wang, Feng Shan and Tong Zhang” is added.
36. In revised supplementary information, page 1, line 14-15, “Joint International Research Laboratory of Information Display and Visualization, School of Electronic Science and Engineering, Southeast University, Nanjing 210096, P. R.” is added.
37. In revised supplementary information, page 1, line 29, “Supplementary sections 1–2” is changed to “Supplementary Sections 1–3”.
38. In revised supplementary information, page 1, line 30, “Supplementary Figures 1–8” is changed to “Supplementary Figures 1-9”.
39. In revised supplementary information, page 2, line 37-62, we have added a section “Section 1. Dissolution of large particles and generation of small ones” to elucidate the chemical and reaction of gold nanoparticles during in-situ observation.
40. In revised supplementary information, page 3, line 74-84, a discussion about dipolar effect is added.
41. In revised supplementary information, page 3, line 87-99, the estimation of Debye screen length is modified by considering other ions.
42. In revised supplementary information, page 4, line 106-124, more details are added to elucidate the extraction of the decay length λ and thickness of ligands L.
43. In revised supplementary information, page 5, line 159, “Figure 4 and S8” is changed to “Figure 4 and S9”.
44. In revised supplementary information, page 5, line 160, “facets” is changed by “surfaces”.
45. In revised supplementary information, page 5, line 163, “all the three facets” is changed to “both these two surfaces”.
46. In revised supplementary information, page 10, line 288-309, Supplementary Figure 4 is modified to provide more information about particle morphology.

47. In revised supplementary information, page 11, line 316, “facets” is changed to “surfaces”
48. In revised supplementary information, page 11, line 316-317, “Both alignment of their {111} and {100} facets lead to a final single crystal structure” is changed to “The contact at their aligned {111} surfaces with parallel {100} facets lead to a final single crystal structure.”
49. In revised supplementary information, page 12, line 328-332, Supplementary Figure 7 is added.
50. In revised supplementary information, page 12, line 336-341. Supplementary Figure 8 is modified to show the oriented attachment at {111} surfaces.

Reviewers' comments:

Reviewer #1 (Remarks to the Author):

The authors have sufficiently addressed the reviewers' comments in their response.

Additional comments:

Figure R1, 2, 3 should be included in supporting materials.

Reviewer #2 (Remarks to the Author):

The manuscript has been well improved. Nonetheless, in my opinion, it had better to highlight the differences between this work and previous one (Science, 2012, 336, 1011), which was specified in rebuttal letter rather than in revised manuscript. For instance, provide useful Figure R4 and the correspondingly text into the manuscript or Supporting Information.

Reviewer #3 (Remarks to the Author):

In the revised manuscript, the authors have improved their work substantially. Regarding the dipolar interaction, it is not safe to state that their effect is subtle. As two attaching crystals go close enough, dipole distribution is expected to be re-arranged and careful discussion over the dipole evolution in the OA growth should thus be presented in the article. Theoretical simulations can be conducted to show the evolution of dipolar interaction in the OA growth and there are means to realize such simulations.

In the meantime, the authors should introduce a few relevant, important recent references in the Introduction, including:

1. A Unified Description of Attachment-Based Crystal Growth
2. An insight into the Coulombic interaction in the dynamic growth of oriented-attachment nanorods
3. Aggregation, Coarsening, and Phase Transformation in ZnS Nanoparticles Studied by Molecular Dynamics Simulations
4. Understanding the oriented-attachment growth of nanocrystals from an energy point of view: a review

Response to the reviewers' comments

Reviewer #1 (Remarks to the Author):

The authors have sufficiently addressed the reviewers' comments in their response.

We thank this referee for her/his positive feedback of our response.

Comment 1: Figure R1, 2, 3 should be included in supporting materials.

Authors' Reply:

We have included the Figure R1, 2, 3 in Supplementary Information (Supplementary Information, page 10, line 309-315; page 15, line 437-442; page 16, line 452-457).

Reviewer #2 (Remarks to the Author):

The manuscript has been well improved.

We thank this referee for her/his positive appreciation of our revised manuscript.

Comment 1: Nonetheless, in my opinion, it had better to highlight the differences between this work and previous one (Science, 2012, 336, 1011), which was specified in rebuttal letter rather than in revised manuscript. For instance, provide useful Figure R4 and the correspondingly text into the manuscript or Supporting Information.

Authors' Reply:

We have included Figure R4 and the discussion in Supplementary Information (Supplementary Information, page 20, line 537-540; page 8, line 238-269). We also have highlighted the differences between this work and previous one in revised manuscript (page 8-9, line 290-295).

Reviewer #3 (Remarks to the Author):

In the revised manuscript, the authors have improved their work substantially.

We thank this referee for her/his positive appreciation of our revised manuscript.

Comment 1: Regarding the dipolar interaction, it is not safe to state that their effect is subtle. As two attaching crystals go close enough, dipole distribution is expected to be re-arranged and careful discussion over the dipole evolution in the OA growth should thus be presented in the article. Theoretical simulations can be conducted to show the evolution of dipolar interaction in the OA growth and there are means to realize such simulations.

Authors' Reply:

We thank this referee for pointing out some facts that we ignored in our analysis. For those particles possessing significant permanent dipoles due to non-centrosymmetric positive and negative charges such as ZnO, PbSe, CdS and etc, dipolar interaction may always have a direct impact on OA process regardless of their separation distance. When it comes to our case, the experiment results show that the gold nanoparticles are highly symmetrical in geometry morphology (please see Supplementary Figure 5). Hence it is reasonable to speculate that ligands are equally adsorbed on the opposite surfaces with same crystal plane index. As a result, an individual capped particle exhibits little dipole moments and weak polarizabilities despite of the existence of some possible local dipole moments. On this basis we did not take dipolar interaction into account in the previous version. However as mentioned by this referee, when the particle get close enough, their surface charges may undergo redistribution due to the mutual influence, and thus leading to re-arranged distribution of dipoles moments. Following the referee's suggestion, we have simulated theoretical model (two attaching Au₇₉ particles with ligands at different distance), and calculated the Bader charges of the model to qualitatively analyze the dipolar distribution during the OA growth process.

In our calculation, considering the complexity of the system and large amount of calculation, the model has been simplified to the configuration that two ligands (C₆H₇O₇) symmetrically

adsorbed on {111} surfaces of Au₇₉ particles. As shown in Figure R1, two citrate ligands symmetrically adsorbed on {111} surfaces of Au₇₉ particles with a size of 1.3 nm.

Figure R1. Configuration of citrate ligands symmetrically adsorbed on {111} surfaces of Au₇₉ particles. **a**, simulation of relaxation of multifaceted Au₇₉ particles with a size of 1.3 nm. **b**, two ligands (C₆H₇O₇) symmetrically adsorbed on {111} surfaces of Au₇₉ particles. Atoms are noted with colors (yellow: gold; red: oxygen; gray: carbon; white: hydrogen).

When the attaching Au particles get closer (Figure R2a), to a distance of about 1.5, 1.3 and 0.9 nm (0.9 nm is close enough in our situation. This is because when the distance reaches about 0.7 nm, the directional rotation stops and citrates detach from surfaces; and more importantly at this moment, the expulsion of surface ligands creates a transient vacuum state between the interspace of two attaching surfaces, making the pressure of surrounding liquid become the dominant force.), the Bader charges of Au atoms between surface 1 and 2 were calculated (Figure R2b), by which the evolution of dipole interaction between Au surfaces is qualitatively evaluated.

Figure R2. Configuration of two attaching Au₇₉ particles with symmetrically adsorbed citrate ligands (at ~1.5 nm). **a**, simulation of relaxation of two attaching Au₇₉ particles with ligands. **b**, **c** ligands (C₆H₇O₇) adsorbed on {111} faces of surfaces 1 and 2, respectively. Atoms are noted with colors (yellow: gold; red: oxygen; gray: carbon; white: hydrogen).

As shown in Table R1, when the separation distance changes from about 1.5, 1.3 and 0.9 nm, the change of Bader charge value of every atom is at most 0.1% for surface 1 and 2.3% for surface 2. It means that when the particles get closer, there is little change in the Bader charges of Au atoms on both surfaces. In other words, as the attaching crystals get close, the redistribution of dipole moments which origin from surface charges could be ignored. The results are summarized here: (1) extremely small permanent dipole moments for individual particle; (2) nearly no redistribution of dipole moments as the particles approach. We can deduce that dipole evolution rarely plays a role in OA growth. Therefore, the contribution of dipolar interaction to OA process is extremely weak in our case, which is also in good accordance with the conclusion from previous experimental work. (*Science*, 2006, 314, 274-278)

Bader charge of Au atoms on surface 1 {111}												
D (nm)	1	2	5	6	7	8	9	14	15	30	33	34
1.5	11.056	11.050	11.053	10.829	10.875	10.836	11.008	11.038	10.987	11.067	11.030	10.967
1.3	11.056	11.051	11.053	10.830	10.875	10.837	11.007	11.039	10.987	11.068	11.031	10.968
0.9	11.056	11.051	11.051	10.829	10.874	10.835	11.007	11.043	10.989	11.064	11.041	10.965

Bader charge of Au atoms on surface 2 {111}												
D (nm)	80	81	84	85	86	87	88	93	94	109	112	113
1.5	11.057	11.046	11.051	10.830	10.875	10.850	11.002	11.034	10.992	11.053	11.032	10.979
1.3	11.055	11.046	11.050	10.804	10.887	10.847	11.005	11.035	10.993	11.061	11.032	10.985
0.9	10.803	10.991	11.037	10.968	10.839	10.937	10.942	11.054	10.996	11.047	11.067	11.047

Table R1. The Bader charges of two attaching Au atoms between surface 1 and surface 2.

We have included above calculation results and discussion in Supplementary Information (page 3, line 82-102; page 7, line 217-220; page 19, line 515-522; page 21, line 560-563). Please let us know if this referee has any other suggestions about the calculation.

Comment 2: In the meantime, the authors should introduce a few relevant, important recent references in the Introduction, including:

1. A Unified Description of Attachment-Based Crystal Growth
2. An insight into the Coulombic interaction in the dynamic growth of oriented-attachment nanorods
3. Aggregation, Coarsening, and Phase Transformation in ZnS Nanoparticles Studied by Molecular Dynamics Simulations
4. Understanding the oriented-attachment growth of nanocrystals from an energy point of view: a review

Authors' Reply:

We have included these references in the Introduction part.

Summary of the changes:

(Changes in the revised manuscript are highlighted in blue color)

1. In revised manuscript, page 1, line 4-5, the authors “Erhong Song and Jianjun Liu” are added because they carried out theoretical calculation during the revision of manuscript.
2. In revised manuscript, page 1, line 11-13, “The State Key Laboratory of High Performance Ceramics and Superfine microstructure, Shanghai Institute of Ceramics, Chinese Academy of Sciences, Shanghai 200050, P. R. China.” is added.
3. In revised manuscript, page 3, line 100, “Supplementary Fig. 1” is corrected to “Supplementary Fig. 1 and 2”.
4. In revised manuscript, page 3, line 108, “Supplementary Fig. 2” is corrected to “Supplementary Fig. 3”.
5. In revised manuscript, page 4, line 128-129, “Supplementary Fig. 4” is corrected to “Supplementary Fig. 5”.
6. In revised manuscript, page 4, line 143, “Supplementary Fig. 5” is corrected to “Supplementary Fig. 6”.
7. In revised manuscript, page 4, line 144, “Supplementary Fig. 6” is corrected to “Supplementary Fig. 7”.
8. In revised manuscript, page 5, line 164-167, “Moreover, according to our observation, this critical distance is always around 1.3 nm and independent of the particle sizes (Supplementary Fig. 5)” is modified to “Moreover, according to more experimental results, this critical distance at which directional rotation begins is always around 1.3 nm and independent of the particle sizes (Supplementary Fig. 8 and 9).”.
9. In revised manuscript, page 6, line 186-187, “The dipolar interaction is also evaluated and its influence is ruled out in our case (Supplementary Fig. 13 and Section 2)” is added.
10. In revised manuscript, page 6, line 195, “Supplementary Fig. 7” is corrected to “Supplementary Fig. 10”.
11. In revised manuscript, page 7, line 226, “Supplementary Fig. 8” is corrected to “Supplementary Fig. 11”.
12. In revised manuscript, page 7, line 239 and 244, “Supplementary Fig. 4” is corrected to “Supplementary Fig. 5”.
13. In revised manuscript, page 8, line 260, “Supplementary Fig. 9” is corrected to “Supplementary Fig. 12”.
14. In revised manuscript, page 8-9, line 290-295, A description is added to show the mechanism difference between this work and previous one.

15. In revised manuscript, page 10, line 357-362, 367-368, some references are added.
16. In revised supplementary information, page 1, line 4-5, the authors “Erhong Song and Jianjun Liu” are added.
17. In revised supplementary information, page 1, line 12-13, “The State Key Laboratory of High Performance Ceramics and Superfine microstructure, Shanghai Institute of Ceramics, Chinese Academy of Sciences, Shanghai 200050, P. R. China.” is added.
18. In revised supplementary information, page 3-4, line 82-102, a discussion is added to evaluate to contribution of dipole moments during OA.
19. In revised supplementary information, page 7, line 217-220, the calculation method of Bader charges is added.
20. In revised supplementary information, page 8, line 238-269, a discussion about the comparison of OA trajectories between our results and other studies is added.
21. In revised supplementary information, page 10, line 309-315, Supplementary Figure 2 is added.
22. In revised supplementary information, page 15, line 437-442, Supplementary Figure 8 is added.
23. In revised supplementary information, page 16, line 452-457, Supplementary Figure 9 is added.
24. In revised supplementary information, page 19, line 515-522, Supplementary Figure 13 is added.
25. In revised supplementary information, page 20, line 537-540, Supplementary Figure 14 is added.
26. In revised supplementary information, page 21, line 560-563, Supplementary Table 1 is added.

REVIEWERS' COMMENTS:

Reviewer #3 (Remarks to the Author):

The authors have addressed the reviewer's comments and the work is acceptable in its current form.